# A Nonlinear Nonlocal Thermoelasticity Euler–Bernoulli Beam Theory and Its Application to Single-Walled Carbon Nanotubes

**DOI:** 10.3390/nano13040721

**Published:** 2023-02-14

**Authors:** Kun Huang, Wei Xu

**Affiliations:** Department of Engineering Mechanics, Faculty of Civil Engineering and Mechanics, Kunming University of Science and Technology, Kunming 650500, China

**Keywords:** Euler–Bernoulli beam theory, nonlocal elasticity, temperature, single-walled carbon nanotubes, buckling, nonlinear vibration

## Abstract

Although small-scale effect or thermal stress significantly impact the mechanical properties of nanobeams, their combined effects and the temperature dependence of the elastic parameters have yet to attract the attention of researchers. In the present paper, we propose a new nonlocal nonlinear Euler–Bernoulli theory to model the mechanical properties of nanobeams. We considered the small-scale effect, thermal stress, and the temperature dependence of Young’s modulus. A single-walled carbon nanotube (SWCNT) was used to demonstrate the influence of the three factors on elastic buckling and forced bending vibrations. The results indicate that thermal stress and the temperature dependence of Young’s modulus have a remarkable influence on the mechanical properties of slender SWCNTs as compared to the small-scale effect induced by the nonlocal effect. Ignoring the temperature effect of slender SWCNTs may cause qualitative mistakes.

## 1. Introduction

Nanobeams have significant potential for application in nanoelectromechanical systems (NEMS) [1,2]. However, how to precisely model the mechanical properties of nanobeams under combined physical fields is still an open research question [3,4]. There are two crucial characteristics in the mechanical properties of nanostructures. One is the small-scale effect, which reveals the dependency between the mechanical properties and the geometric size of nanostructures [5,6,7,8]. The other is the ambient temperature that may soften the stiffness of nanobeams [3,9,10,11,12]. The small-scale effect [13,14,15] and thermal stress [3,16] of nanobeams have been extensively studied. However, researchers have rarely paid attention to the change in the elastic parameters induced by temperature for the nonlinear vibrations of nanobeams. Nanobeams are usually used at room temperature. Therefore, clarifying the mechanical characteristics under small-scale effect and a finite temperature is required.

The nonlocal stress gradient theory has been used to model the small-scale effect [17,18]. In this theory, the relationship between the local stress and the nonlocal stress is $\left[1-{\left(e_{0}a\right)}^{2}\nabla^{2}\right]\sigma_{ij}={\bar{\sigma }}_{ij}$ [17]. Here, ${\bar{\sigma }}_{ij}$ is the local stress, $\sigma_{ij}$ is the nonlocal stress, $e_{0}$ is a small-scale parameter, and a is the material characteristic length, which is the carbon–carbon bonding length for SWCNTs. The stress gradient model has been widely used to describe the mechanical properties of CNTs and graphene [19,20,21]. As the mechanical properties of a single atomic layer require a more thorough understanding based on quantum mechanics [22,23], determining the scale parameter $e_{0}$ remains a controversial open question [24].

A change in temperature may affect the properties of nanomaterials by inducing thermal stress and changing the elastic parameters [11,12,25,26,27]. There is much research on the thermal stress of nanobeams [3,9,11], but the dynamic effects of temperature dependence on the elastic parameters have been scarcely considered by researchers [8,28,29]. The temperature dependence in macro structures cannot be ignored arbitrarily [27]. Because the stiffness of nanobeams is much smaller than that of macrobeams, they are more sensitive to temperature changes. In addition, whether the coupling of temperature and scale effect significantly affects the mechanical properties of nanobeams is also an important research content. Material nonlinearity may impact the nonlinear vibrations of CNTs [6,30]. This influence decreases with an increase in the length of CNTs [6].

The accurate understanding of the mechanical properties of nanostructures is the basis of applications. Comprehensively considering the small-scale effect and the temperature is necessary. Therefore, in the present study, we propose a new beam theory to include the nonlocal effect and temperature.

## 2. Materials and Methods

We restricted our attention to slender beams to remove the influence of material nonlinearity [6]. Thus, the Euler–Bernoulli hypothesis was employed [31]. The cross-sections perpendicular to the centroid locus before deformation remained on the plane and perpendicular to the deformed locus, suffering no strain on their planes. Under this hypothesis, only the longitudinal (x-direction) strain component of the beam ${\bar{\varepsilon }}_{xx}$ was considered (Figure 1). For simplicity, we assumed that the strain of the beam is finite but small. The local longitudinal stress can be expressed as ${\bar{\sigma }}_{xx}={\bar{\sigma }}_{xx}^{0}+\bar{E}{\bar{\varepsilon }}_{xx}$ [31]. Here, ${\bar{\sigma }}_{xx}^{0}$ is the local stress before displacement and $\bar{E}$ is the elasticity modulus at the ambient temperature. If a beam has two unmovable ends, we have ${\bar{\sigma }}_{xx}^{0}={\bar{\sigma }}_{0}+\bar{E}\gamma T$. Here, γ is the coefficient of thermal expansion (CTE) [25] and ${\bar{\sigma }}_{0}$ is the initial prestress. Recent studies have shown that the elasticity modulus $\bar{E}$ of CNTs proportionally decreases with an increase in temperature [10,11,32]. As the elasticity modulus was obtained using molecular dynamic simulations at absolute zero, we modelled this linear relationship of CNTs as $\bar{E}=E\left(1-\gamma_{1}T\right)$. Here, $\gamma_{1}>0$ is the coefficient of thermoelasticity expansion (CEE) and E is the elasticity modulus at absolute zero. This formula indicates the softening behavior of the tubes with the temperature. Following the nonlocal differential constitutive relationship [17], a constitutive relationship of the nanobeam with the temperature effect can be written as:(1)$$\left[1-\mu^{2}\nabla^{2}\right]\sigma_{xx}={\bar{\sigma }}_{xx}={\bar{\sigma }}_{xx}^{0}+\bar{E}{\bar{\varepsilon }}_{xx}=-{\bar{\sigma }}_{0}-E\left(1-\gamma_{1}T\right)\gamma T+E\left(1-\gamma_{1}T\right){\bar{\varepsilon }}_{xx},$$
here, $\mu =e_{0}a$.

The bending may induce the axial extension under finite deformations, for example, hinged–hinged or clamped–clamped beams. We suppose that u and w are axial displacements of a beam in the x- and *y*-directions, respectively, as shown in Figure 1. Thus, the axial strain is [6,31]:(2)$${\bar{\varepsilon }}_{xx}=\frac{\partial u}{\partial x}+\frac{1}{2}{\left(\frac{\partial w}{\partial x}\right)}^{2}-y\frac{\partial^{2}w}{\partial x^{2}}$$

Substituting Equation (2) into the local stress ${\bar{\sigma }}_{xx}$, we have the local axial force and bending moment as:(3)$$\bar{N}=\int_{A}{\bar{\sigma }}_{xx}dA=-N_{0}-AE\left(1-\gamma_{1}T\right)\gamma T+AE\left(1-\gamma_{1}T\right)\left[\frac{\partial u}{\partial x}+\frac{1}{2}{\left(\frac{\partial w}{\partial x}\right)}^{2}\right],$$
(4)$$\bar{M}=\int_{A}E\left(1-\gamma_{1}T\right)y{\bar{\sigma }}_{xx}dA=-\left(1-\gamma_{1}T\right)EI\frac{\partial^{2}w}{\partial x^{2}}.$$

In the equations, A and $I=\iint_{A}^{}y^{2}dydz$ are the cross-sectional area and the area moment of inertia of the cross-section. $N_{0}=\sigma_{0}A$ is the initial pretension force, as shown in Figure 1. The equations of motion with the extensional effect are [31]:(5)$$\frac{\partial N}{\partial x}=m\frac{\partial^{2}u}{\partial t^{2}},$$
(6)$$\frac{\partial^{2}M}{\partial x^{2}}+N\frac{\partial^{2}w}{\partial x^{2}}=m\frac{\partial^{2}w}{\partial t^{2}}+\bar{F}\left(x,t\right).$$

Here, m is the mass per unit length of the beam. N and M are the nonlocal axial force and the nonlocal moment, respectively. From Equations (3) and (4), the nonlocal constitutive Equation (1) can be rewritten as:(7)$$M-\mu^{2}\frac{\partial^{2}M}{\partial x^{2}}=-E\left(1-\gamma_{1}T\right)I\frac{\partial^{2}w}{\partial x^{2}},$$
(8)$$N-\mu^{2}\frac{\partial^{2}N}{\partial x^{2}}=-\left[N_{0}+AE\left(1-\gamma_{1}T\right)\gamma T\right]+\left(1-\gamma_{1}T\right)AE\left[\frac{\partial u}{\partial x}+\frac{1}{2}{\left(\frac{\partial w}{\partial x}\right)}^{2}\right].$$

Substituting Equations (5) and (6) into Equations (7) and (8), we obtain:(9)$$M-\mu^{2}\left(m\frac{\partial^{2}w}{\partial t^{2}}-N\frac{\partial^{2}w}{\partial x^{2}}+\bar{F}\right)=-\left(1-\gamma_{1}T\right)EI\frac{\partial^{2}w}{\partial x^{2}},$$
(10)$$N-\mu^{2}\frac{\partial }{\partial x}\left(m\frac{\partial^{2}u}{\partial t^{2}}\right)=-N_{0}-A\left(1-\gamma_{1}T\right)\gamma T+\left(1-\gamma_{1}T\right)EA\left[\frac{\partial u}{\partial x}+\frac{1}{2}{\left(\frac{\partial w}{\partial x}\right)}^{2}\right],$$

We differentiate Equation (9) twice with respect to x. We then substitute it into Equation (6) to obtain:(11)$$-\left(1-\gamma_{1}T\right)EI\frac{\partial^{4}w}{\partial x^{4}}+N\frac{\partial^{2}w}{\partial x^{2}}+\mu^{2}\frac{\partial^{2}}{\partial x^{2}}\left(m\frac{\partial^{2}w}{\partial t^{2}}-N\frac{\partial^{2}w}{\partial x^{2}}+\bar{F}\right)\;=m\frac{\partial^{2}w}{\partial t^{2}}+\bar{F}\left(x,t\right).$$

From Equation (10), we obtain:(12)$$\begin{array}{c} N=\mu^{2}\frac{\partial }{\partial x}\left(m\frac{\partial^{2}u}{\partial t^{2}}\right)-\left[N_{0}+AE\left(1-\gamma_{1}T\right)\gamma T\right]+\left(1-\gamma_{1}T\right)EA\left[\frac{\partial u}{\partial x}+\frac{1}{2}{\left(\frac{\partial w}{\partial x}\right)}^{2}\right]. \end{array}$$

Substituting Equation (12) into Equation (11), we obtain:(13)$$-\left(1-\gamma_{1}T\right)EI\frac{\partial^{4}w}{\partial x^{4}}-\left[N_{0}+AE\left(1-\gamma_{1}T\right)\gamma T\right]\frac{\partial^{2}w}{\partial x^{2}}+\left(1-\gamma_{1}T\right)EA\left[\frac{\partial u}{\partial x}\right.\;\left.+\frac{1}{2}{\left(\frac{\partial w}{\partial x}\right)}^{2}\right]\frac{\partial^{2}w}{\partial x^{2}}-\mu^{2}\frac{\partial^{2}}{\partial x^{2}}\left\{-\left[N_{0}+AE\left(1-\gamma_{1}T\right)\gamma T\right]\frac{\partial^{2}w}{\partial x^{2}}\right.\;\left.\left.+\left(1-\gamma_{1}T\right)EA\left[\frac{\partial u}{\partial x}+\frac{1}{2}{\left(\frac{\partial w}{\partial x}\right)}^{2}\right]\frac{\partial^{2}w}{\partial x^{2}}\right]\right\}=m\left(\frac{\partial^{2}w}{\partial t^{2}}-\mu^{2}\frac{\partial^{4}w}{\partial t^{2}\partial x^{2}}\right)+\bar{F}-\mu^{2}\frac{\partial^{2}\bar{F}}{\partial x^{2}}.$$

We ignored the inertia term $\partial^{2}u/\partial t^{2}$ in Equation (13) because it is significantly smaller than $\partial^{2}w/\partial t^{2}$ for a slender beam [33]. We differentiated Equation (12) with respect to x and then substituted it into Equation (5) to obtain:(14)$$\left(1-\gamma_{1}T\right)EA\frac{\partial }{\partial x}\left[\frac{\partial u}{\partial x}+\frac{1}{2}{\left(\frac{\partial w}{\partial x}\right)}^{2}\right]=m\frac{\partial^{2}u}{\partial t^{2}}-\mu^{2}m\frac{\partial^{4}u}{\partial t^{2}\partial x^{2}}.$$

Equations (13) and (14) are the plane motion equations for the nanobeam with the nonlocal effect and the temperature. If only the bending motion is considered, we can ignore the inertia terms of Equation (14) for a slender beam [33] to obtain:(15)$$\frac{\partial^{2}u}{\partial x^{2}}=-\frac{1}{2}\frac{\partial }{\partial x}{\left(\frac{\partial w}{\partial x}\right)}^{2}$$

Integrating Equation (15) with respect to x, we obtain:(16)$$\frac{\partial u}{\partial x}=-\frac{1}{2}{\left(\frac{\partial w}{\partial x}\right)}^{2}+C_{1}\left(t\right),\;u=-\frac{1}{2}\int_{0}^{x}{\left(\frac{\partial w}{\partial s}\right)}^{2}ds+C_{1}\left(t\right)x+C_{2}\left(t\right)$$
where $C_{1}$ and $C_{2}$ are functions of time t because u is a function of t and x. The two parameters can be determined by imposing boundary conditions on w. For a beam with two unmovable ends [33],
(17)$$C_{1}\left(t\right)=\frac{1}{2l}\int_{0}^{l}{\left(\frac{\partial w}{\partial x}\right)}^{2}dx,\;\;C_{2}=0.$$

By substituting Equation (16) into Equation (13) and omitting the quartic terms of w, we obtain:(18)$$m\left(\frac{\partial^{2}w}{\partial t^{2}}-\mu^{2}\frac{\partial^{4}w}{\partial t^{2}\partial x^{2}}\right)+C\frac{\partial w}{\partial t}+\left[N_{0}+AE\left(1-\gamma_{1}T\right)\gamma T\right]\frac{\partial^{2}w}{\partial x^{2}}\;\;\;\;\;\;\;\;\;\;\;+\left\{\left(1-\gamma_{1}T\right)EI-\mu^{2}\left[N_{0}+AE\left(1-\gamma_{1}T\right)\gamma T\right]\right\}\frac{\partial^{4}w}{\partial x^{4}}\;\;\;\;-\frac{\left(1-\gamma_{1}T\right)EA}{2l}\left[\frac{\partial^{2}w}{\partial x^{2}}\right.\left.-\mu^{2}\frac{\partial^{4}w}{\partial x^{4}}\right]\int_{0}^{l}{\left(\frac{\partial w}{\partial x}\right)}^{2}dx=\bar{F}+\mu^{2}\frac{\partial^{2}\bar{F}}{\partial x^{2}}.$$

Equation (18) indicates that the initial pretension has an identical effect to the thermal stress. Thus, we neglected the initial pretension $N_{0}$ for simplicity. We assumed that the transverse load is uniform, namely $\bar{F}\left(x\right)=const$. Thus, $\partial^{2}\bar{F}/\partial x^{2}=0$. In this case, Equation (18) can be rewritten as:(19)$$m\left(\frac{\partial^{2}w}{\partial t^{2}}-\mu^{2}\frac{\partial^{4}w}{\partial t^{2}\partial x^{2}}\right)+C\frac{\partial w}{\partial t}+\left(I-\gamma \mu^{2}AT\right)E\left(1-\gamma_{1}T\right)\frac{\partial^{4}w}{\partial x^{4}}\;+AE\left(1-\gamma_{1}T\right)\gamma T\frac{\partial^{2}w}{\partial x^{2}}-\frac{\left(1-\gamma_{1}T\right)EA}{2l}\left[\frac{\partial^{2}w}{\partial x^{2}}\right.\left.-\mu^{2}\frac{\partial^{4}w}{\partial x^{4}}\right]\int_{0}^{l}{\left(\frac{\partial w}{\partial x}\right)}^{2}dx=\bar{F}.$$

From Equation (19), we observed two key features. First, the temperature softens the stiffness of the nanobeams if γ>0 and $\gamma_{1}>0$. Second, the new model will degenerate as the classical beam theory including the thermal stress if $\gamma_{1}=\mu^{2}=0$. We added a linear damping term C∂w/∂t to Equation (19) to account for the dissipation of energy. For a hinged–hinged beam, the boundary conditions are [31,33]:(20)$$w\left(0,t\right)=w\left(l,t\right)=0,\;\frac{\partial^{2}w}{\partial x^{2}}\left(0,t\right)=\frac{\partial^{2}w}{\partial x^{2}}\left(l,t\right)=0.$$

From the structural point of view, SWCNTs can be thought of as a single sheet of graphene, rolled into a cylindrical shape with axial symmetry. The ‘rolling up’ of the graphene sheet is described by the chiral vector, which can be expressed as (*m*,*n*), where integers *n* and *m* represent the chiral indices. An SWCNT is armchair if *m* = *n*. For a tube with thickness h and middle surface diameter d, the area moment of inertia of the cross-section is $I=\frac{\pi }{64}\left[{\left(d+h\right)}^{4}-{\left(d-h\right)}^{4}\right]=\frac{\pi hd}{8}\left(d^{2}+h^{2}\right)$. Thus, the bending stiffness is $EI=E\pi dh\left(d^{2}+h^{2}\right)/8$. The tensile stiffness is EA=Eπdh, and $EI/EA=\left(d^{2}+h^{2}\right)/8$. Several studies have shown that the two stiffnesses are independent for SWCNTs [3,34]. However, the stiffnesses of SWCNTs with a large diameter are close to those of thin-walled circular tubes. In the present study, a $\left(10,10\right)$ SWCNT is used as an example to demonstrate the mechanical properties of the nanobeams. The diameter is d=1.356 nm, h=0.34 nm, $m=2.238\times 10^{-15}\;\mathrm{kg}/m$, and $E=1.086\times 10^{3}\;\mathrm{GPa}$ at absolute zero [35]. The other parameters are shown in Table 1. The CTE was obtained from [12] and the CEE was obtained by fitting the data of the MD calculations from [11].

## 3. Results

Introducing dimensionless variables into Equation (19) is convenient. Let $\bar{x}=x/l$, $\bar{w}=w/l$, $\bar{t}=\omega_{0}t$, and $\omega_{0}^{2}=\pi^{4}EI{\left(l^{4}m\right)}^{-1}$. Equation (19) can then be rewritten as
(21)$$\frac{\partial^{2}}{\partial {\bar{t}}^{2}}\left(\bar{w}-\frac{\mu^{2}}{l^{2}}\frac{\partial^{2}\bar{w}}{\partial {\bar{x}}^{2}}\right)+\frac{C}{m\omega_{0}}\frac{\partial \bar{w}}{\partial t}+\left(\frac{1-\beta }{\pi^{4}}-\frac{\beta_{1}\mu^{2}}{l^{2}}\right)\frac{\partial^{4}\bar{w}}{\partial {\bar{x}}^{4}}\;+\beta_{1}\frac{\partial^{2}\bar{w}}{\partial {\bar{x}}^{2}}-\beta_{2}\left(\frac{\partial^{2}\bar{w}}{\partial {\bar{x}}^{2}}\right.\left.-\frac{\mu^{2}}{l^{2}}\frac{\partial^{4}\bar{w}}{\partial {\bar{x}}^{4}}\right)\int_{0}^{1}{\left(\frac{\partial \bar{w}}{\partial \bar{x}}\right)}^{2}d\bar{s}=\frac{\bar{F}}{m\omega_{0}^{2}l}.$$

The parameters in Equation (21) are
(22)$$\beta =\frac{\gamma_{1}TEI+\left(1-\gamma_{1}T\right)\gamma \mu^{2}EAT}{m\omega_{0}^{2}l^{2}},\;\beta_{1}=\frac{\left(1-\gamma_{1}T\right)\gamma TAE}{m\omega_{0}^{2}l^{2}},\;\beta_{2}=\frac{\left(1-\gamma_{1}T\right)EA}{2m\omega_{0}^{2}l^{2}}.$$

The normalized boundary conditions are
(23)$$\bar{w}\left(0,\bar{t}\right)=\bar{w}\left(1,\bar{t}\right)=0,\;\;\frac{\partial^{2}\bar{w}}{\partial {\bar{x}}^{2}}\left(0,\bar{t}\right)=\frac{\partial^{2}\bar{w}}{\partial {\bar{x}}^{2}}\left(1,\bar{t}\right)=0$$

It is challenging to accurately solve nonlinear Equation (21). An approximate method to solve nonlinear equations is to reduce the partial differential equation to nonlinear ordinary differential equations and then solve the equations using perturbation methods [33,36]. Another method is to directly solve the nonlinear partial differential equation using the multiscale method [37,38,39]. We applied the Galerkin and the multiscale methods to approximately solve the equation. As Equation (21) and the classical nonlinear beam theory have the same form apart from the coefficients in the equations, their modes in the Galerkin method were identical. Under the boundary conditions of Equations (23), an approximate solution to Equation (21) is
(24)$$\bar{w}=\sum_{n=1}^{\infty }{\bar{\eta }}_{n}\left(\bar{t}\right)\sin n\pi \bar{x}$$

Ordinary differential equations were obtained through the Galerkin truncation [31,33]. We substituted Equation (24) into Equation (21); then, $\sin\left(n\pi \bar{x}\right)$ was multiplied by both sides of the equation and integrated into the interval $\left[0,1\right]$. For simplicity, we only took the first term of Equation (24) and let ${\bar{\eta }}_{1}=\eta$ to obtain
(25)$$\bar{m}\ddot{\eta }+\bar{C}\dot{\eta }+k\eta +d\eta^{3}=F.$$

The parameters in Equation (25) are
(26)$$\bar{m}=1+\frac{\pi^{2}\mu^{2}}{l^{2}},\;\;\bar{C}=\frac{C}{m\omega_{0}},\;\;k=1-\beta -\frac{\pi^{4}\mu^{2}\beta_{1}}{l^{2}}-\pi^{2}\beta_{1},\;d=\frac{\beta_{2}}{2}\left(\pi^{4}+\frac{\pi^{6}\mu^{2}}{l^{2}}\right),\;\;F=\frac{4\bar{F}}{\pi m\omega_{0}^{2}l}.$$

Equations (26) indicate that the linear and the nonlinear stiffness parameters are functions of the nonlocal parameters and temperature. We used a $\left(10,10\right)$ SWCNT to demonstrate the influence of the nonlocal effect and the temperature on the nonlinear mechanical properties. The parameters are shown in Table 1. Determining the nonlocal parameter $e_{0}$ of SWCNTs is an open problem because there is an ambiguous understanding of the mechanical properties of a one-atom-thick nanostructure [22,23]. If the vibration frequency in CNTs is in the terahertz range, a conservative estimate is $e_{0}a<2\;\mathrm{nm}$ [8]. In this research, $e_{0}a=0.6\;\mathrm{nm}$ [13] was used.

By neglecting the inertia and damping terms in Equation (24), static bending deformations of the middle point of the beam are obtained as
(27)$$k\eta +d\eta^{3}=F.$$

From the classical theory, the beam will buckle if k=0. We can also obtain post-buckling displacements with the temperature for different parameters, as shown in the next section.

To research the nonlinear vibrational behaviors of a hinged–hinged beam at the primary resonance, we rewrite Equation (25) as
(28)$$\ddot{\eta }+\bar{c}\dot{\eta }+{\bar{\omega }}^{2}\eta +\bar{d}\eta^{3}=\bar{f}$$
where $\bar{c}=\bar{C}/\bar{m}$, $\omega^{2}=k/\bar{m}$, $\bar{d}=d/\bar{m}$, and $\bar{f}=F/\bar{m}$. The energy dissipation of nanostructures is not thoroughly understood; thus, we used $\bar{c}=0.04$ as an example. The multiple scale method [36], which is widely used to solve weak nonlinear differential equations for macro-structures, was applied to solve Equation (28). Let $\bar{c}=2\varepsilon^{2}c$, $\bar{f}=\varepsilon^{3}f\cos\left(\Omega \bar{t}\right)$ and ε=0.1. This gives
(29)$$\ddot{\eta }+\omega^{2}\eta +2\varepsilon^{2}c\dot{\eta }+\bar{d}\eta^{3}=\varepsilon^{3}f\cos\left(\Omega \bar{t}\right).$$

Suppose that
(30)$$\eta \left(\bar{t};\varepsilon \right)=\varepsilon \eta_{1}\left(T_{0},T_{2}\right)+\varepsilon^{3}\eta_{3}\left(T_{0},T_{2}\right)$$
where $T_{0}=\bar{t}$ and $T_{2}=\varepsilon^{2}\bar{t}$. We substitute Equation (30) into Equation (29), and then equate the coefficients of ε and $\varepsilon^{3}$ on both sides. This gives
(31)$$\varepsilon :\;D_{0}^{2}\eta_{1}+\omega^{2}\eta_{1}=0,$$
(32)$$\varepsilon^{3}:\;D_{0}^{2}\eta_{3}+\omega^{2}\eta_{3}=-2D_{0}D_{2}\eta_{1}-2cD_{0}\eta_{1}-\bar{d}\eta_{1}^{3}+\frac{1}{2}f\exp\left(i\Omega T_{0}\right)$$
in which $D_{0}=d/dT_{0}$, $D_{2}=d/dT_{2}$, and $D_{0}^{2}=d^{2}/dT_{0}^{2}$. The solution of Equation (31) is
(33)$$\eta_{1}=A\left(T_{2}\right)\exp\left(i\omega T_{0}\right)+CC$$
where *CC* is the complex conjugate. Substituting Equation (33) into Equation (32), we obtain
(34)$$D_{0}^{2}\eta_{3}+\omega^{2}\eta_{3}=\frac{1}{2}f\exp\left(i\omega T_{0}\right)-\left[i\;2\omega \left(A^{\prime }+cA\right)\right.\left.+3\bar{d}A^{2}\bar{A}\right]\exp\left(i\omega T_{0}\right)+NST$$

The prime in Equation (34) denotes the derivative with respect to $T_{2}$ and NST denotes the non-secular terms [36]. When the frequency Ω of the load approaches the modal frequency ω (primary resonance) of the nanobeam, a small-amplitude excitation may produce a relatively large amplitude response. Under this condition, let $\Omega =\omega +\varepsilon^{2}\sigma$; thus, the solvable condition of Equation (34) is
(35)$$-i2\omega \left(A^{\prime }+cA\right)-3\bar{d}A^{2}\bar{A}+\frac{1}{2}f\exp\left(i\sigma T_{2}\right)=0$$

Let $A=\left(\alpha /2\right)\exp\left(i\mu \right)$, and substitute it into Equation (35). We then separate the real part and the imaginary part to obtain
(36)$$\alpha^{\prime }=-c\alpha +\frac{f}{2\omega }\sin\theta ,\;\alpha \theta^{\prime }=\sigma \alpha -\frac{3}{8\omega }\bar{d}\alpha^{3}+\frac{f}{2\omega }\cos\theta ,$$
where $\theta =\sigma T_{2}-\mu$. Steady-state motion occurs when $\alpha^{\prime }=\theta^{\prime }=0$, which corresponds to the singular points of Equations (36). The steady-state solution can be obtained from the following algebraic equation [36]
(37)$$\left[c^{2}+{\left(\sigma -\frac{3\bar{d}}{8\omega^{2}}\alpha^{2}\right)}^{2}\right]\alpha^{2}=\frac{f^{2}}{4\omega^{2}}$$

The stability of the steady-state solution is judged by investigating the nature of the singular points of Equations (38). Let $a=a_{0}+a_{1}$, $\theta =\theta_{0}+\theta_{1}$, and substitute them into Equation (36), expanding for small $a_{1}$ and $\theta_{1}$, and maintaining the linear terms in $a_{1}$ and $\theta_{1}$. We then obtain
(38)$$\alpha_{1}^{\prime }=-c\alpha_{1}+\frac{\theta_{1}f\cos\theta_{0}}{2\omega },\;\theta_{1}{}^{\prime }=-\left(\frac{f\cos\theta_{0}}{2\omega \alpha_{0}^{2}}+\frac{3\bar{d}\alpha_{0}}{4}\right)\alpha_{1}-\frac{\theta_{1}f\sin\theta_{0}}{2\omega \alpha_{0}}.$$

Here, $a_{0}$ and $\theta_{0}$ are the singular points of Equation (36) that are used in Equation (38). The stability of the steady-state motion depends on the eigenvalues of the coefficient matrix of Equation (36). If the real parts of the eigenvalues are greater than zero, the solutions are unstable [36]. Hence, the steady-state motions are unstable if
(39)$$c^{2}+\left(\frac{3\bar{d}\alpha_{0}^{2}}{8\omega }-\sigma \right)\left(\frac{9\bar{d}\alpha_{0}^{2}}{8\omega }-\sigma \right)<0$$

Equation (39) indicates that the nonlocal effect and the temperature impact the stability of the steady-state solutions. Therefore, we used dashed and solid lines to denote the unstable and stable solutions, respectively, in the figures of the following section.

## 4. Discussion

If F=0 in Equation (27), one can obtain the buckled deflections induced by temperature in different models, as shown in Figure 2. The image indicates that the nonlocal effect strikingly impacted the buckling temperature because there are a few coupled terms of $\gamma_{1}$ in the stiffness k. This means that it is necessary to consider the influence of the temperature on the elastic parameter. The temperature effect on the bending deformations is mainly displayed by the softening of the stiffness due to γ>0 and $\gamma_{1}>0$ for SWCNTs, as shown in Figure 3. This figure also demonstrates that the beam has been buckled by thermal stress at T=600 K, considering the nonlocal and temperature effects.

From Equation (37), we obtained the effect of the temperature on the load–amplitude curves, as shown in Figure 4. The numerical calculations using the Runge–Kutta method showed the accuracy of the perturbation solution, as shown in Figure 5. From Equations (21) and (25), we observed that the nonlocal effect and the temperature induced a few terms. Figure 4 shows that these coupled terms were the reason why the vibrations of the SWCNT were sensible to the temperature. The nonlocal effect or the temperature-induced decrease in Young’s modulus had little influence on the vibration amplitudes of the SWCNTs at the same temperature, as shown in Figure 6. However, the three models had different bifurcation points that induced a jump in the vibration amplitudes. The sensibility of the mechanical properties to the temperature could also be found in the frequency–response curves, as shown in Figure 7. Figure 7 and Figure 8 also indicate that qualitative mistakes appeared if the temperature was ignored. The frequency–response curves for the three models, as shown in Figure 9, also demonstrated that the nonlocal effect or Young’s modulus change had a minor influence on the vibration amplitudes at the same temperature.

The preliminary studies demonstrated that the temperature significantly affected the vibration behavior of the SWCNTs. However, we did not discuss their oscillations after buckling induced by temperature. The dynamic behaviors of the buckling tubes were more complicated than those in this paper [33]. Particularly, the bifurcation problem with two parameters requires further theoretical and experimental research. Our new model provides a basis to study these problems profoundly.

The correctness of the new theory in this paper requires experimental validation. However, experiments of mechanical properties at the nanoscale are very difficult, and those of nonlinear vibrations are even more difficult. The molecular dynamics calculations of the nonlinear vibrations are equally difficult due to enormous expenses. To date, there are no experimental data of nonlinear vibrations of CNTs under finite temperature conditions. It is challenging to obtain the nonlinear vibrations of CNTs using the molecular dynamics calculations. In the present research, we use the mechanical and thermal coefficients obtained using the density functional theory or molecular dynamics simulations. This ensures accuracy of the model parameters. Recent molecular dynamics calculations of SWCNTs show that the buckling deformations of CTs under axial compression loads is highly similar to those of classical beams [40]. This enhances the researchers’ confidence in using the modified continuum mechanics to study nanobeams [6,8,41]. We believe that the nonlinear mechanical properties of nanobeams require further theoretical and experimental research.

## 5. Conclusions

In the present study, we combined the nonlocal effect and temperature to suggest a new Euler–Bernoulli theory for nanobeams. A $\left(10,10\right)$ SWCNT was used as an example to demonstrate the combined impact of the nonlocal effect, the thermal stress, and the temperature dependence of Young’s modulus. The research demonstrated three main impacts induced by temperature:(1).The nonlocal effect and the decrease in Young’s modulus substantially influenced the buckling temperature and the post-buckling mechanical behavior.(2).The temperature had a softening effect on the stiffness of the SWCNTs and remarkably impacted the nonlinear vibrations of the structures. These effects increased with an increase in the temperature.(3).As both the nonlocal effect and the temperature effects significantly impacted the mechanical properties of SWCNTs, noticeable mistakes appeared if they were neglected in the model.

## Figures and Tables

**Figure 1 nanomaterials-13-00721-f001:**
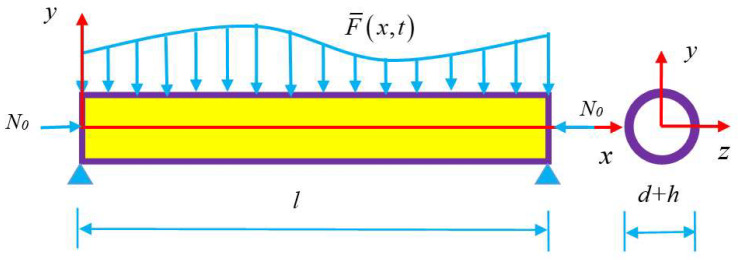
Schematic configuration of a thin-walled nanobeam with the middle surface diameter d and the thickness h of the wall.

**Figure 2 nanomaterials-13-00721-f002:**
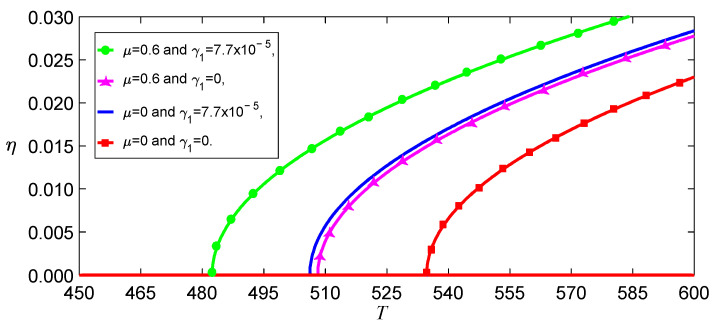
Buckled deflections induced by temperature in different models for F=0.

**Figure 3 nanomaterials-13-00721-f003:**
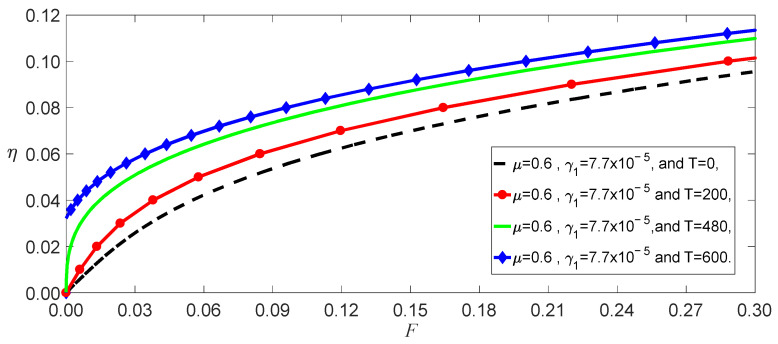
Displacements as a function of loads for different temperatures.

**Figure 4 nanomaterials-13-00721-f004:**
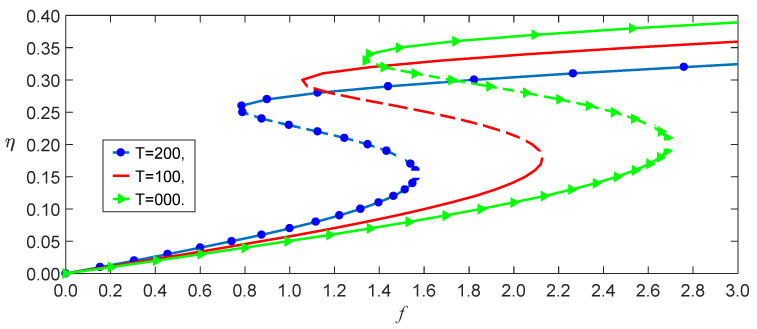
The temperature effect on the load–response curves for $\left(\sigma ,\mu ,\lambda_{1}\right)=\left(10,\;0.6,\;7.7\times 10^{-5}\right)$.

**Figure 5 nanomaterials-13-00721-f005:**
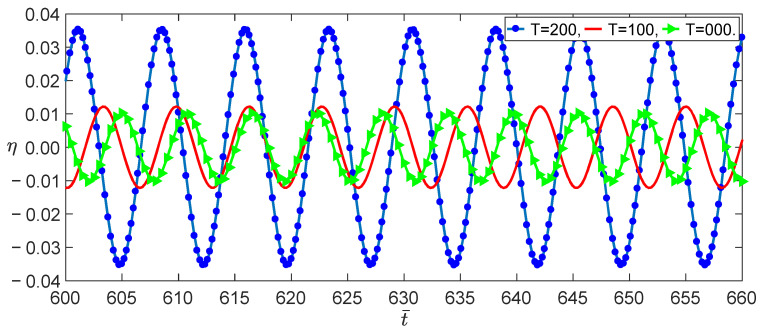
Time evolutions with three different temperatures for f=0.002 at an initial value of $\left(\eta_{0},{\dot{\eta }}_{0}\right)=\left(0,0\right)$.

**Figure 6 nanomaterials-13-00721-f006:**
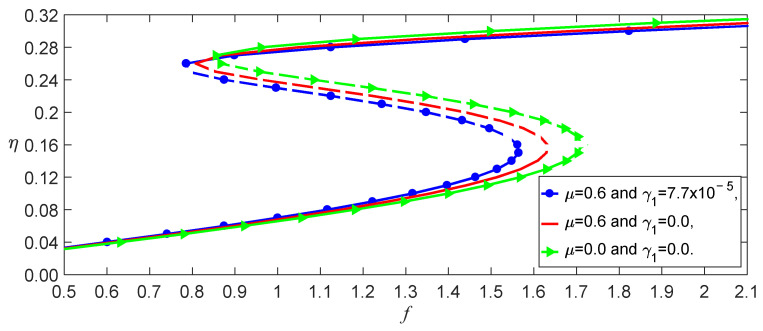
Load–response curves of the three models for $\left(T,\sigma \right)=\left(200,10\right)$.

**Figure 7 nanomaterials-13-00721-f007:**
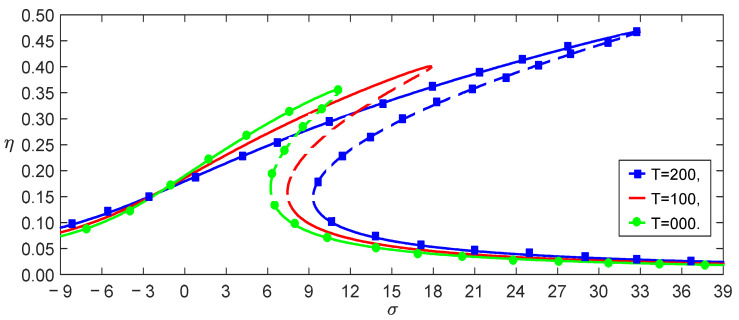
Frequency–response curves with three temperatures for f=1.5.

**Figure 8 nanomaterials-13-00721-f008:**
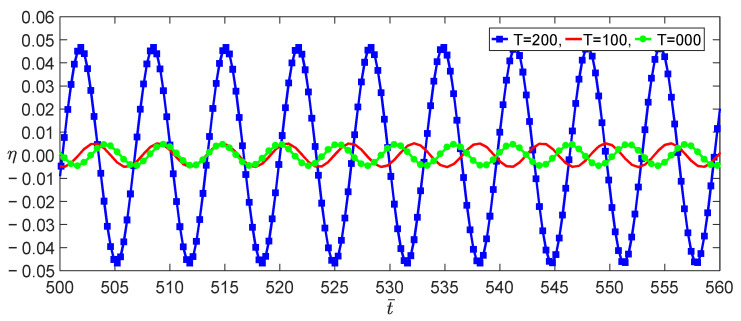
Time evolutions with three temperatures for $\left(\sigma ,f\right)=\left(20,1.5\right)$ at the initial value of $\left(\eta ,\dot{\eta }\right)=\left(0.1,0\right)$.

**Figure 9 nanomaterials-13-00721-f009:**
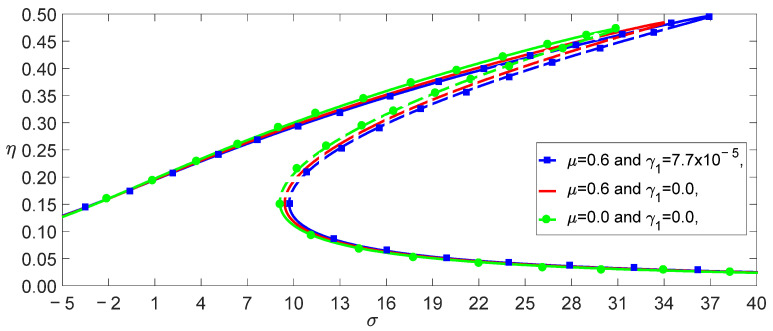
Frequency–response curves of three models for $\left(f,T\right)=\left(1.5,\;200\right)$.

**Table 1 nanomaterials-13-00721-t001:** The physical parameters of a $\left(10,10\right)$ SWCNT with l=15 nm.

Bending Rigidity ($\mathrm{nN}\cdot {\mathrm{nm}}^{2}$)	Extensional Rigidity (nN)	γ ($T^{-1}$)	$\gamma_{1}$ ($T^{-1}$)	$\omega_{0}^{2}$	μ (nM)	$\bar{c}$
EI=384.07	EA=1572.16	$2\times 10^{-5}$	$7.7\times 10^{-5}$	$2.277\times 10^{23}$	0.6	0.04

## Data Availability

The datasets generated during and/or analyzed in the current study are available from the corresponding author on reasonable request.

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
