# Peer review of "A Nonlinear Nonlocal Thermoelasticity Euler–Bernoulli Beam Theory and Its Application to Single-Walled Carbon Nanotubes"

_nanomaterials, 2023, doi:10.3390/nano13040721_

Round 1
Reviewer 1 Report
The authors study the temperature-dependent nonlinear mechanical property of SWCNT in a given geometry. Here, they suggest a generalization of the Euler-Bernoulli theory. The aim seems acceptable.
My criticisms and comments refer to the calculations in the article. I see several typos and missing definitions. These are sometimes extremely confusing for the reader. E.g.:
In Eq. (3): Instead of the factor A(E-\gamma_1 T) -- > AE (1-\gamma_1 T)
In line 85: There is no definition for the inertia moment I. The unit of it is m^4, so it requires more explanation.
In Eq. (6): The appearance of the first term must be shortly justified. It is not enough to cite a more than 800 pages book.
The authors ignore the inertia term in line 102 (before Eq. (13)). However, the inertia term appears again in Eq. (14). They ignore it again in Eq. (15). The time dependence appears again in Eq. (16). I understand the purpose, but it's hard to follow.
In line 140: \overline{t} = t/\omega_0 is wrong, or \overline{t} is not dimensionless. \overline{t} = t * \omega_0
It is not clear what is going on after Eq. (29). Cos(\Omega \overline{t}). \overline{t} is dimensionless, so \Omega is also dimensionless. In line 206: \Omega = \omega + \varepsilon^2 \sigma, in line 187: \omega^2 = k/m is not dimensionless. Please, clarify. What is the definition of \varepsilon?
What is the meaning of Eq. (30). What are T_0 and T_2?
What are D_0 and D_2 in Eqs (31) and (32)? In line 199: D_1, but I do not see D_2 here.
\gamma is the coefficient of the thermal expansion in line 63. What is \gamma in Eq. (36)? The parameter \beta in line 208 is not the \beta in Eq. (22).
These confusing errors need to be corrected.
Author Response
Dear reviewer:
Thank you for your comments. These comments help us to improve the quality of the manuscript. We revised the manuscript according to the reviewers’ comments, and carefully proofread the manuscript to minimize typographical, grammatical, and bibliographical errors. Significant changes have been marked in red font. Here below is our description of the revision according to your comments.
Sincerely yours.
Kun Huang
2023/01/31
Major Remarks:
Comment (1): The authors study the temperature-dependent nonlinear mechanical property of SWCNT in a given geometry. Here, they suggest a generalization of the Euler-Bernoulli theory. The aim seems acceptable.
My criticisms and comments refer to the calculations in the article. I see several typos and missing definitions. These are sometimes extremely confusing for the reader. E.g.:
In Eq. (3): Instead of the factor A(E-\gamma_1 T) -- > AE (1-\gamma_1 T)
Reply: Thank you for your comments. These comments helped us to improve the quality of the manuscript.
We thank you for pointing out the typing error. We have corrected this error in the revised version. Furthermore, we revised the manuscript and carefully proofread the manuscript to minimize typographical, grammatical, and bibliographical errors. Significant changes have been marked in red font.
Comment (2): In line 85: There is no definition for the inertia moment I. The unit of it is m^4, so it requires more explanation.
Reply: In this paper, the area moment of inertia of the cross-section is . For a tube with the thickness and the middle surface diameter , it is . In the revision, we have explained this issue at the lines 93-94 and 141-143 in red.
Comment (3): In Eq. (6): The appearance of the first term must be shortly justified. It is not enough to cite a more than 800 pages book.
Reply: We mainly cite the beam theory in the literature [31] at pages 286-317. We have added page numbers after Ref. [31].
Comment (4): The authors ignore the inertia term in line 102 (before Eq. (13)). However, the inertia term appears again in Eq. (14). They ignore it again in Eq. (15). The time dependence appears again in Eq. (16). I understand the purpose, but it's hard to follow.
Reply: We have ignored the inertia term of u in Eq. (13) because it is far less than that of w for a slender beam. Because Eq. (14) describes the longitudinal vibrations, the inertia term induced by the nonlocal effect is retained. This inertia term is necessary for the longitudinal vibrations. The inertia term of u is omitted from Eqs. (15) and (16) in order to derive Eq. (18) in simple form. This treatment is feasible in classical beam theory, as shown in Ref. [31]. We have added this issue on lines 113-124 in red in the version.
Comment (5) and (6): In line 140: \overline{t} = t/\omega_0 is wrong, or \overline{t} is not dimensionless. \overline{t} = t * \omega_0
It is not clear what is going on after Eq. (29). Cos(\Omega \overline{t}). \overline{t} is dimensionless, so \Omega is also dimensionless. In line 206: \Omega = \omega + \varepsilon^2 \sigma, in line 187: \omega^2 = k/m is not dimensionless. Please, clarify. What is the definition of \varepsilon?
Reply: There is a typographical error when we introduce dimensionless variables in lines 155-156. The correct form is: \overline{t} = t*\omega_0. By correcting an error, the misunderstanding in Eq. (29) can be eliminated. We have corrected the above error in the revised version. Thank you again for pointing out our negligence.
Comment (7) and (8): What is the meaning of Eq. (30). What are T_0 and T_2?
What are D_0 and D_2 in Eqs (31) and (32)? In line 199: D_1, but I do not see D_2 here.
Reply: T_0=\overline{t} and T_2=\varepsilon^2 \ overline{t}. D_0=d/D(T_0) and D_2=d/d(T_2). We mistakenly write D_2 as D_1. We have revised the above two questions in lines 214-219 of the revised version.
Comment (9): \gamma is the coefficient of the thermal expansion in line 63. What is \gamma in Eq. (36)? The parameter \beta in line 208 is not the \beta in Eq. (22).
Reply: We replaced \gamma with \theta and replaced \beta with \mu in Eqs. (36) and (38) in order to eliminate the confusion. These modifications are indicated at lines 230-242 in red.
Reviewer 2 Report
In this paper, a nonlocal nonlinear Euler-Bernoulli model is addressed. Thermal stress and the dependence temperature of Young’s modulus are considered. Elastic buckling and forced vibrations are investigated.
The paper deserves attention.
Some comments follow.
- Authors should clear address the actual novelty of the paper in view of similar works in the literature as well as versus authors' previous works.
- Please, check English style and typos (e.g. "The nonlocal stress gradient theory have used to model the small-scale effect").
- Last sentence in the abstract "Moreover, it may cause qualitative mistakes to ignore the temperature effect of slender SWCNTs." should be better clarified. Do authors mean the thermal stresses or the young's modulus variation versus temperature? The first one is quite obvious and well known.
- Validation is missing and it is crucial for the paper to be accepted since without it, the presented results are all questionable.
- "If two ends of a beam cannot move along the x -axis, such as hinged-hinged or clamped-clamped beams, the bending deformations may induce the axial extension". You have also axial strain in a cantilever where an end is free. In general, bending will result in, at least, an axial strain.
- What are omega^2_0 and overbar c in table 1? They should be defined when addressing table 1 in the text.
- Eq. 23: displacements and derivatives cannot be equated even if both are zero since are different quantities.
- The quality of some pictures (e.g., Fig. 1 and 2) is very poor. Please, replace them with better ones.
- Sections 3 and 4 can be merged.
Author Response
Dear reviewer:
Thank you for your comments. These comments helped us to improve the quality of the manuscript.. We revised the manuscript according to the reviewers’ comments, and carefully proofread the manuscript to minimize typographical, grammatical, and bibliographical errors. Significant changes have been marked in red font. Here below is our description of the revision according to your comments.
Sincerely yours.
Kun Huang
2023/01/31
Comment (1): In this paper, a nonlocal nonlinear Euler-Bernoulli model is addressed. Thermal stress and the dependence temperature of Young’s modulus are considered. Elastic buckling and forced vibrations are investigated.
The paper deserves attention.
Some comments follow.
- Authors should clear address the actual novelty of the paper in view of similar works in the literature as well as versus authors' previous works.
Reply: Thank you for your comments. These comments helped us to improve the quality of the manuscript.
The main innovation of this paper is that demonstrates the elastic parameter’s change induced by temperature strongly affects the mechanical behavior of SWCNTs, and this effect even exceed the nonlocal scale effect. This has not been paid attention to by researchers before. We have added the preamble to the revised version along lines 48-52 in red.
Comment (2): Please, check English style and typos (e.g. "The nonlocal stress gradient theory have used to model the small-scale effect").
Reply: We revised the manuscript according to the reviewers’ comments, and carefully proofread the manuscript to minimize typographical, grammatical, and bibliographical errors. Moreover, the revised version has been professionally edited in English at MDPI.
Comment (3): Validation is missing and it is crucial for the paper to be accepted since without it, the presented results are all questionable.
Reply: It is challenging to obtain the mechanical, thermal properties of CNTs by experiments. For mechanical and thermal expansion properties, the available experimental data are consistent with the results of quantum calculations. However, there are not experimental data of nonlinear vibrations under finite temperature conditions so far. Therefore, in the present research, we use the mechanical and thermal coefficients obtained by the density functional theory or molecular dynamics simulations. This ensures accuracy of the model parameters. In fact, it is very difficult to obtain the nonlinear vibration behaviors of nanobeams or CNTs under finite temperature conditions through atomic calculation.
Of course, the correctness of the theory requires experimental check. We are not experts in experimental nanophysics, and we do not have the capability of experimental studies. We believe that the mechanical properties of nanobeams require further theoretical and experimental research. For the response to this problem, we have provided additional clarification in lines 273-279 of the version in red.
Comment (4): "If two ends of a beam cannot move along the x -axis, such as hinged-hinged or clamped-clamped beams, the bending deformations may induce the axial extension". You have also axial strain in a cantilever where an end is free. In general, bending will result in, at least, an axial strain.
Reply: We modified this expression as “The bending may induce the axial extension under finite deformations, for example hinged-hinged or clamped-clamped beams.” in lines 84-85 in red.
Comment (5): What are omega^2_0 and overbar c in table 1? They should be defined when addressing table 1 in the text.
Reply: \omega^2_0 is the square of the frequency that are used in of dimensionless time, as shown in line 156 in red. “overbar c” is damping parameters, as shown in Eq. (28) and line 205 in red.
Comment (6): Eq. 23: displacements and derivatives cannot be equated even if both are zero since are different quantities.
Reply: We have given their respective expressions in the revised draft in Eqs. (23) and (20).
Comment (7): The quality of some pictures (e.g., Fig. 1 and 2) is very poor. Please, replace them with better ones.
Reply: We replaced these two figures with better quality figures in the revised version.
Comment (8): Sections 3 and 4 can be merged.
Reply: The current version of this paper is in the format required by the submission journal.
Reviewer 3 Report
The authors propose a nonlocal
nonlinear Euler-Bernoulli theory to model single-walled carbon nanotube by
considering the effect the small-scale effect, the thermal stress, and the
dependence temperature of Young’s modulus. Nonlinear terms are considered in
deformation. The study is applied to simply supported nanotube beam where
thermic stresses are present due to restrained axial supports. Nonlinear equilibrium equation are present in
buckling and in the motion equation in dynamics where Duffing’s equation is
obtained and solved by the multiscale method.
The manuscript must be strengthened according to the journal standards and templates and other major comments that must be solved. They are following:
- The mechanical properties change at high temperature depending on the material under study. Here, in the Eq of line 66, the authors consider changes for any temperature amplitude T. In general, the reference temperature is taken at room value.
- What means d of the nanotube. It must be clear in Fig 1: define d and h separately. In general, the diameter of the tube is defined for the external parts, the thickness is the wall dimension. The authors have to check the equations of nanotube cross section parameters A and I in Line 127 (proofs)
- There is no comparisons to benchmark solutions or other numerical simulations. The topic is not new !
- The model is semi analytical applied strictly to simply supported conditions according to admitted mode Eq. 24. What about the other possible boundary conditions in beam theory
Others:
- Line 44: ‘The ambient temperature affects nanomaterials by inducing thermal stress and changing elastic parameters’ This equation is not clear: Ambiant or high temperature.
- Presence of many important equation in the text. In general, the equations must be in separate line with an equation number : eqs in line 37, 61, 63, 66, 127…
- The title are figs and table are not clear. Ex
Lin 180: ‘Figure. 2 The displacement as a function of temperature for different models with
This title is not explicit, since it represents the buckling and post-buckling behaviour of the nanotube in terms of and
- Same in Fig 3
- what relation of (10.10) in Line 131 and (10,10) in the title of table1
- The quality of some figs are not good (Fig1 and 2)
- Line 146: ‘It is challenging to solve the nonlinear Eq. (23) accurately. ‘ :But this eq is a boundary cond, the solution is known. Check : Perhaps Eq 21?
- Check the terms of Line 199, Define D2
- Line 210: define c1 in Eq 36?
- Give the units of TPa for a non familiar with the topic in relation to MPa, GPa.
English
Line 35 ‘The nonlocal stress gradient theory have used to model…
Line 37 ‘ , Here’ : after a comma, write ‘, where ij is …’ This phrase has repeated many times in the text, as:
Line 61: Here, xx is the stress
Line 63 : Here is the coefficient
Line 72: here e 0a
Line 90 : Here m is the beam’s
…
Line 91,92: ‘the non-local constitutive Eq. (1) can rewrite as:
Line 11 A beam with two unmovable ends has[33].
Write as ‘A beam with two unmovable ends has [33] .
Line 123: 1 not used in the model (perhaps 1?)
Line 205: ‘ the beam will appear a relatively large amplitude response.’

Author Response
Dear reviewer:
Thank you for your comments. These comments helped us to improve the quality of the manuscript.. We revised the manuscript according to the reviewers’ comments, and carefully proofread the manuscript to minimize typographical, grammatical, and bibliographical errors. Significant changes have been marked in red font. Here below is our description of the revision according to your comments.
Sincerely yours.
Kun Huang
2023/01/31
Comment (1):
The authors propose a nonlocal nonlinear Euler-Bernoulli theory to model single-walled carbon nanotube by considering the effect the small-scale effect, the thermal stress, and the dependence temperature of Young’s modulus. Nonlinear terms are considered in deformation. The study is applied to simply supported nanotube beam where thermic stresses are present due to restrained axial supports. Nonlinear equilibrium equation are present in buckling and in the motion equation in dynamics where Duffing’s equation is obtained and solved by the multiscale method.
The manuscript must be strengthened according to the journal standards and templates and other major comments that must be solved. They are following:
The mechanical properties change at high temperature depending on the material under study. Here, in the Eq of line 66, the authors consider changes for any temperature amplitude T. In general, the reference temperature is taken at room value.
Reply: Thank you for your comments. These comments helped us to improve the quality of the manuscript.
Because the elasticity modulus is usually obtained using molecular dynamic Simulations at absolute zero, we take the temperature T as the absolute temperature. This make us can directly use the data from atomic calculations. We explain this issue in lines 71-74 of the revised version in red.
Comment (2): What means d of the nanotube. It must be clear in Fig 1: define d and h separately. In general, the diameter of the tube is defined for the external parts, the thickness is the wall dimension. The authors have to check the equations of nanotube cross section parameters A and I in Line 127 (proofs)
Reply: “d” is the middle surface diameter of the tube, and “h” is the thickness in Fig. 1. We have explained the two parameters in the title of Fig. 1 in the revised version. We check the formulas for I and A, and they are correct. We have supplemented for the two parameters on lines 141-143 of the revised version in red.
Comment (3): There is no comparisons to benchmark solutions or other numerical simulations. The topic is not new !
Reply: Of course, the correctness of the theory requires experimental check. It is challenging to obtain the mechanical, thermal properties of CNTs by experiments. However, there are not experimental data of nonlinear vibrations under finite temperature conditions so far. Therefore, in the present research, we use the mechanical and thermal coefficients obtained by the density functional theory or molecular dynamics simulations. This ensures accuracy of the model parameters. In fact, it is also very difficult to obtain the nonlinear vibration behaviors of nanobeams or CNTs under finite temperature conditions through atomic calculation.
We are not experts in experimental nanophysics, and we do not have the capability of experimental studies. We believe that the mechanical properties of nanobeams require further theoretical and experimental research. For the response to this problem, we have provided additional clarification in lines 273-279 of the version in red.
Comment (4): The model is semi analytical applied strictly to simply supported conditions according to admitted mode Eq. 24. What about the other possible boundary conditions in beam theory.
Reply: In this paper, we only studied the nonlinear mechanical behavior of carbon nanotubes through a hinge-hinge beam. For other boundary conditions, the Galerkin’s modes of the beam are different from these of a hinge-hinge beam. But the resulting ordinary differential equations have some slight difference in the equation coefficients. This makes their conclusions with different boundary conditions is qualitatively consistent. Of course, a careful comparison between different boundaries is necessary. But, this will greatly increase the length of the paper. It might be more appropriate to clarify these complex issues in another paper.
Comment Others:
- Line 44: ‘The ambient temperature affects nanomaterials by inducing thermal stress and changing elastic parameters’ This equation is not clear: Ambiant or high temperature.
- Presence of many important equation in the text. In general, the equations must be in separate line with an equation number : eqs in line 37, 61, 63, 66, 127…
3.The title are figs and table are not clear. Ex
Lin 180: ‘Figure. 2 The displacement as a function of temperature for different models with
This title is not explicit, since it represents the buckling and post-buckling behaviour of the nanotube in terms of and
- Same in Fig 3
- what relation of (10.10) in Line 131 and (10,10) in the title of table1
- The quality of some figs are not good (Fig1 and 2)
- Line 146: ‘It is challenging to solve the nonlinear Eq. (23) accurately. ‘ :But this eq is a boundary cond, the solution is known. Check : Perhaps Eq 21?
7 Check the terms of Line 199, Define D2
8 Line 210: define c1 in Eq 36?
9 Give the units of TPa for a non familiar with the topic in relation to MPa, GPa.
Reply: The questions above are answered below in order of title:
- This paper takes the modulus of elasticity at absolute zero as the benchmark, so the present study focus on that the temperature is greater than absolute zero. To clarify the confusion, we have revised the expression of lines 44 and 72-74 in red.
- The expression in lines 62-75 is mainly to obtain the Eq.(1). This paper only cites Eq. (1)in subsequent discussions. In the revised version, we have rewritten equation (1) to better express the content between 62-75 lines.
- We rewrite the headings of Fig.s 2 and 3, as well as lines 194-204 in the revised version. This allows the content of the diagram to be better interpreted.
- (10.10) in Line 146 is an error. it is (10,10) and is same as Table.1.
- We replaced these two figures with better quality figures in the revised version.
- This is typing error. The equation should be Eq. (21). We have corrected this error in the revised version.
- We mistakenly write D_2 as D_1. We have revised the above two questions in lines 207-218 of the revised version.
- This is typing error. “c1” should be “c”. We have corrected this error in the revised version.
- We have changed "TPa" to "10^3GPa" in the revised version.
Comment English:
Line 35 ‘The nonlocal stress gradient theory have used to model…
Line 37 ‘ , Here’ : after a comma, write ‘, where ij is …’ This phrase has repeated many times in the text, as:
Line 61: Here, xx is the stress
Line 63 : Here is the coefficient
Line 72: here e 0a
Line 90 : Here m is the beam’s
…
Line 91,92: ‘the non-local constitutive Eq. (1) can rewrite as:
Line 11 A beam with two unmovable ends has[33].
Write as ‘A beam with two unmovable ends has [33] .
Line 123: 1 not used in the model (perhaps 1?)
Line 205: ‘ the beam will appear a relatively large amplitude response
Reply: We revised the manuscript according to the reviewers’ comments, and carefully proofread the manuscript to minimize typographical, grammatical, and bibliographical errors. The issues mentioned above have been modified and marked in red. Moreover, the revised version has been professionally edited in English at MDPI.
Round 2
Reviewer 1 Report
The authors revised the manuscript. I recommend the acceptance of it for publication.
Author Response
Dear reviewer:
Thank you so much. We revised the manuscript again, and carefully proofread the manuscript to minimize typographical, grammatical, and bibliographical errors. Significant changes have been marked in red font.
Sincerely yours.
Kun Huang
2023/02/07
Reviewer 2 Report
The paper in its current form is supported for publication.
Author Response

(The authors gave the same response as above.)

Reviewer 3 Report
Comments are enclosed in the pdf file.

Author Response
Dear reviewer:
Thank you for your comments. These comments helped us to improve the quality of the manuscript. We revised the manuscript according to your comments, and carefully proofread the manuscript to minimize typographical, grammatical, and bibliographical errors. Significant changes have been marked in red font. Here below is our description of the revision according to your comments.
Sincerely yours.
Kun Huang
2023/02/07
Comment (1): The work is a semi analytical method strictly applied for simply supported beam in bending. According to the present version, one remark many problems present. This means that authors must be careful before submitting the future version. Many basic problems are remarked in the model that get the results more doubtful for the scientific community. The major problems are the following:
- To my opinion, the lines 155 until 194 are parts of the semi-analytic method developed in section 2. Before The application section, authors must develop, the static equation in a dimensional and after the dynamic equation by including inertia and damping and the solution procedure for each behavior (statics, dynamics).
- According to Eq 25, (in both statics and dynamic), it is believed to solve the equation in statics (m, C=0) and after in dynamics. In statics, inertia terms are omitted and the force term F in 26 is not true and must modified according to static condition
- Same eq 27 same reason.
Reply: This paper focuses on the influence of temperature on the mechanical properties of nanobeams under the nonlocal stress-strain relations. We then used single-walled carbon nanotubes as an example to demonstrate this effect. Experiments of mechanical properties at the nanoscale are very difficult, and these of nonlinear vibrations are even more difficult. In fact, the molecular dynamics calculations of nanobeam’s nonlinear vibrations are equally difficult. Due to the lack of reference, how to test the theoretical results of this paper requires further study. Using a solution procedure to solve static and dynamic problems is an optional method to check the results of this paper. In fact, existing literatures have some studies in which temperature effects are not considered. Ones can find references in Ref. [8]. Because the new model in present paper is similar in the models without the influence of temperature on the elastic constant, we can predict that the results of the solution procedure are consistent with the results in this paper for the statics. However, how to calculate nonlinear vibrations using the solution procedure, for example the finite element, is a challenging problem. And adding these content will greatly increase the length of the paper. So we only added a discussion of this problem in lines 284-296 in red. We intend to discuss this issue in another paper.
Because we did dimensionless in Equation (21), Equation (27) is dimensionless after the Galerkin truncation. This makes the load F in Equation (27) be a dimensionless parameter. Although the frequency parameter omega is included in F, it is only a parameter used to nondimensionalize load. The purpose of the dimensionless is to facilitate the perturbation of Equation (28). For these reasons, Equation (27) is suitable for static bending.
Comment (2): Same in Eq 28, again in ref to Eq 26 F=???????? ?? (F,m,l, ω2) and again you define f= F/m so f becomes function of (F,m2, l, ω2) there is a confusing problem ? It is believed to derive the dynamic eq from 25.
- This is one reason that the results have to be considered with more caution since there is no comparison with benchmark solutions. With errors remarked in the model, the results are not confident at all. At minimum, compare your results to available solutions for isotropic beams in buckling and linear vibration. Possible comparions are available here, for buckling vibration
and post-buckling bending behaviour.
DOI: 10.1142/S0219455412500459
DOI 10.21595/jve.2015.16751
DOI 10.1186/s13661-016-0561-3
DOI:10.1016/S0263-8231(01)00038-6
Reply: The coefficients in Equation (28) are obtained by dividing the coefficient of Equation (26) by "m-bar". The manuscript is incorrectly written as m. We have corrected this error in the version. We thank you for pointing out the error.
Thank you for the literatures you recommend. They inspire us a lot. Two of them are related to this paper and may help readers understand progress in nanobeams. So we added them to the bibliography and commented on them (References [40] and [41]). However, because the effects of temperature were not studied in these literatures, we did not compare their results with our conclusions in the revision. We have added discussion in lines 284-296 in red to discuss the detection of the correctness in our paper.
Comment (3): Line 196: This phrase is not clear in ref to Fig 3 and Fig2. In Fig 2. The buckling Temp is around 482° for mu=0.6 and γ1=7.7e-5 and not 600° as in Fig 3 where η begings with 0.04 for F=0?
Reply: Considering the effect of temperature on the modulus of elasticity, CNTs will be buckled at 482. This makes the CNT has a deformation 0.04 induced by temperature at 600 without the external force, as shown in Figure 3. We supplemented some comments at lines 249-257 of the revised version in red.
Comment (4): to be clear between γ and γ1, it is believed to denote the thermal coefficient by α. Be careful adopt another symbolic letter in the dynamic part
Reply: In this article, we used “alpha” for amplitude of vibration. In the nonlinear vibrations, “alpha” is often used to represent the amplitude. There are many symbols in this paper, and in order not to cause more confusion, we have not made further adjustments to the symbols in the revision.
Comment (5): Temperature is varied from 0 to very high temperature (600°). Have you some indication on the melting temperature of the material.
Reply: We used an SWCNT as the research object. Its melt Temperature is 1800 and much more than 600. If one applies the present theory to other materials, it is necessary to consider the melting temperature.
Comment (6): It is believed to represent the beam behaviour with displacement in the horizontal axis and load , or Temperature in the vertical axis, (Figs 2,3,4,6,7..)
Reply: For nonlinear vibrations, it is customary to express deformations on vertical axes. This gives a clearer representation of vibration amplitude jumps. We predict that the potential readers of this article will mostly be in the field of nonlinear dynamics, so we have not adjusted the coordinate representation of the paper.
Comment (7): In general, the tube geometry is defined by the external diameter d and the thickness t or h and not the medium diameter. This is the reason of my previous comment. When d is the medium diameter of the tube A and I are now good.
Reply: We define “d” as the diameter of the middle surface because a single-walled carbon nanotube has not a physical thickness. To make a connection with classical continuum mechanics and the carbon tube, researchers usually define an equivalent thickness “h”. However, the equivalent thickness has different values in different literature. In order to avoid confusion due to differences in equivalent thickness, “d” is usually defined as the diameter of the middle surface.
Comment (8): The integral relationship for I in line 93 is for a continuous cross section and not applied for a tube.
Reply: The moment of inertia of the cross-section at line 93 is the usual definition. For tubes with thickness “h”, this definition can be used to obtain specific expressions at line 150. Because the theory in this paper can be applied not only to tubes, but also to other sections, we give the usual definition of the moment of inertia of the section at line 93.
Comment (9): After Eq 6, respect the equation order as before: the axial eq followed by the bending one as in 5 and 6 and previously.
Reply: Equations (5) and (6) are longitudinal and transverse equations of motion, while equations (7) and (8) are non-local constitutive equations. There is no logical relation between the two sets of equations. We did not reorder the equations in the version for ease of writing.
Comment (10): Some terms in 12 have been omitted in (13) why?
Reply:We ignored the inertia term of u in Equation (13) because it is significantly smaller than that of w for a slender beam. We have explained this problem at lines 114-115 in the revised draft in red.
Comment (11): What means a (10,10) SWCNT used 2 times.
Reply:From the structural point of view, CNTs can be thought of as single sheets of graphite (graphene), rolled into a cylindrical shape with axial symmetry. The ‘rolling up’ of the graphene sheet is described by the chiral vector that can be expressed as (m, n). Where integers n and m represent the chiral indices. (10, 10) indicates m= n=10. This means that we use an armchair SWCNT as example in this paper because the armchair CNTs are isotropic. We have explained this problem at lines 145-148 in the revised draft in red.
Comment (12): The table 1 is not at the good place in the text. It must be in the application part after defining all the terms in the table. One don’t understand the meaning of "c_bar" .
Reply: We have adjusted Table 1 to the front of Equation (29). It will make readers easily understand the paper.
Comment (13): In Lin260: the term bifurcation is not correct in ref to Fig 4 and 6. This term is true in fig 2. Use limit point
Reply: Jump of vibration amplitudes in nonlinear vibrations are caused by bifurcation in the amplitude response. From a mathematical point of view, the bifurcation point is also the limit point of the response curve. But for readers of vibration research, it is more customary to use “bifurcation point”.
Comment (14): Meaning of data 1, in fig.5?
Reply: “data 1” is a typing error, it is “T=0”. Thank you for pointing out our negligence. We have made corrections in the version.
Comment (15): English:
‘We differentiated Equation (12) with respect to x and then substituted it into Equation (5) to obtain:’ write in present time, or like this :’After by differentiating… one obtains:
See other many typos
Reply: We have seriously revised the manuscript, corrected grammatical errors, and improved the writing of the paper.

Round 3
Reviewer 3 Report
Accept as the present form